# Developing a Scene-Based Triangulated Irregular Network (TIN) Technique for Individual Tree Crown Reconstruction with LiDAR Data

**Haijian Liu [1,2] and Changshan Wu [3,*]** 

[1]  Institute of Remote Sensing and Earth Sciences, Hangzhou Normal University, Hangzhou 311121, China; haijian@hznu.edu.cn
[2]  Zhejiang Provincial/Key Laboratory of Urban Wetlands and Regional Change, Hangzhou 311121, China
[3]  Department of Geography, University of Wisconsin-Milwaukee, 3210 N. Maryland Ave, Milwaukee, WI 53211, USA
*   Correspondence: cswu@uwm.edu; Tel.: +414-229-4860; Fax: 414-229-3981

**Abstract:** LiDAR (Light Detection and Ranging)-based individual tree crown reconstruction is a challenge task due to the variable canopy morphologies and the penetrating properties of LiDAR to tree crown surfaces. Traditional methods, including LiDAR-derived rasterization, low-pass filtering smooth algorithm, and original triangular irregular network (TIN) model, have difficulties in balancing morphological accuracy and model smoothness. To address this issue, a scene-based TIN was generated with three steps based on the local scene principle. First, local Delaunay triangles were formed through connecting neighboring point sets. Second, key control points within each local Delaunay triangle, including steeple, inverted tip, ridge, saddle, and horseshoe shape control points, were extracted by analyzing multiple local scenes. These key points were derived to determine the fluctuations of forest canopies. Third, the scene-based TIN model was generated using the control points as nodes. Visual analysis indicates the new model can accurately reconstruct different canopy shapes with a relatively smooth surface, and statistical analysis of individual trees confirms that the overall error of the new model is smaller than others. Especially, the scene-based TIN derived raster reduced the average error to 0.18 m, with a standard deviation of 0.41, while the average errors of LiDAR-derived raster, low-pass filtered smooth raster, and original TIN derived raster have average errors of 0.96, 2.05, and 1.00 m, respectively. The local scene-based control point extraction also reduces data storage due to the elimination of redundant points, and furthermore the different point densities on different objects are beneficial for canopy segmentation.

**Keywords:** scene-based tin; individual tree crown reconstruction; control points; lidar

## 1. Introduction

Individual tree crown reconstruction is essential for forest quality assessment, health supervision, and planning management of forest resources [1]. A three-dimensional (3D) digitization of tree crowns can create virtual models through 3D tools to present more direct spatial information. Similar to 3D buildings, 3D canopy models can be applied to visually represent the morphological structure of forests, as well as quantitatively describe the characteristic parameters of forest topologies [2]. Accurate forest 3D models have shown promise in identifying tree position [3,4], segmenting crown scales [5,6], and estimating vegetation canopy biophysical variables, such as tree heights [7], crown dimensions [8], and biomass [9]. Further, canopy models have been employed as an important means to analyze weather, climate, and air quality [10].

LiDAR (Light Detection and Ranging) has proven to be an effective and increasingly popular technology for 3D model reconstruction of terrains, buildings, transportation facilities, pipelines, and vegetation [11,12]. Small-footprint LiDAR data records dense and discrete 3D points reflected from targeted objects, while full-waveform LiDAR data continuously measures the intensity of returned light over a period of time and acquires a relatively complete waveform profile [13]. Therefore, LiDAR has the advantage of extracting accurate terrain and semantic information [14] and generating forest canopy models from the individual tree level to global level [15]. In the forest application, various algorithms have been proposed for canopy model creation using LiDAR data, which can be grouped into three major categories: rasterized canopy height model (CHM), geometric model, and vectorized mesh model.

CHM is also called normalized digital surface model (nDSM), and is regarded as the difference between the digital surface model (DSM) and the corresponding digital elevation model (DEM), which were usually generated through interpolating the first and last returns of LiDAR data, respectively. Jan [16] applied a CHM model to estimate canopy height in forest stands and found LiDAR performed better than aerial photos in forest canopy structure investigations. Tseng et al. [17] developed a method to map CHM in heterogeneous forests using airborne waveform LiDAR datasets and the CHM estimation is with an estimation error of approximately 0.8 m. Due to the laser penetration ability and steep slopes of terrains, some abnormal changes of elevation exist in the derived CHM regularly [18–20]. In order to accurately characterize the top canopy surface, Ben-Arie et al. [21] developed a semiautomated pit filling algorithm to create smooth CHMs. Popescu and Wynne [22] and Popescu et al. [8] extracted the highest LiDAR elevations in each small cells (0.25 m$^2$) to derive the top DSM with the Kriging interpolation method, and the resulting top DSM was higher than the surface obtained from all first-return LiDAR heights, with an average difference of 0.17 m and the largest difference of 25.19 m. Liu and Dong [3] sorted the LiDAR points within search windows and selected the ones greater than user-defined threshold to create a CHM with inverse distance weighted (IDW) method, and results show the selecting and sorting mechanism outperformed the median filter and highest points method. Khosravipour et al. [19] generated a pit-free CHM raster by using subsets of the LiDAR points to close pits. Although the rasterized model is popular due to the ease of creation, the height of the canopy is underestimated, and the crown shape is skewed [23]. The ordered array of numerical values in the CHM cannot accurately reflect the disordered canopy morphology.

The geometric model reconstructs the canopy surface by optimally fitting the LiDAR points and thus obtains relatively clear canopy vertices, boundaries, or morphologies. Sheng et al. [24] proposed a 3D hemi-ellipsoid model for conifer-crown surface reconstruction. With four parameters including treetop position, crown height, crown radius, and coefficient of crown curvature, the conifer crown can be modeled by a geometric equation. Morsdorf et al. [25] applied a rotational paraboloid model to reconstruct the forest scene for wildland fire management based on the extracted tree height, tree position, crown diameter, and crown base height derived from LiDAR data. Van Leeuwen et al. [23] developed a parametric height model (PHM) to create a series of cones representing the canopy surface of coniferous, from which the negative height bias of LiDAR data can be corrected, and tree crown delineation can be extracted. Liu and Wu [5] created a general geometric model to simulate different crown shapes including cones, semi-sphere, half-ellipsoids, and others, and then tree height and shape-related parameters were extracted as independent variables for a regression model to predict crown widths. Harikumar et al. [26] applied geometric features describing both the internal and external structures of crown to perform conifer species classification. 3D geometric models can vividly simulate the shape of canopy and exhibit the physical characteristics. However, the 3D canopy model is often created at the tree level, and thus issues associated with position determination, boundary recognition, and height estimation limit the wide-ranging applications of forest geometric models.

The triangular irregular network (TIN) model is an alternative to the grid-based model and geometric model as it shows the original shape of objects and predicts the values in an unsampled location [27]. TIN represents the surface by a set of contiguous and nonoverlapping triangles connecting

the original data points [28], thus 3D visualizations are readily created by rendering of triangular facets. TIN mesh has been used to solve many problems including building a topographic map, buffers for objects, and multiplayer information [29]. Specifically, Yang et al. [30] constructed a multiresolution TIN model for visualization. Ali and Mehrabian [31] presented a computational framework to create a TIN for a relatively flat area, based on the graph-theoretical duality between Delaunay triangulation (DT) and Voronoi diagrams. Uysal and Polat [32] filtered out non-ground points from raw point clouds using a TIN algorithm to create digital elevation models in both urban and rural areas. Antonarakis et al. [33] constructed TINs based on elevations and the average intensity of airborne LiDAR data to classify forest and ground land cover types, and five types of riparian forest were classified with accuracies between 66% and 98%. Vuckovik et al. [34] developed a Durkin's propagation model using a TIN-based terrain to assess the performances of ad hoc networks. Zhao et al. [35] proposed an improved progressive TIN densification (IPTD) filtering algorithm to cope with various forested landscapes and achieved higher filtering accuracy when compared to other algorithms. TIN is widely used for terrain modeling, building reconstruction, and transportation analysis [36–38]. However, the application of TIN in forestry is still limited due to the disordered distribution of LiDAR on the surface and interior of forest canopies.

The canopy morphology of forests, especially deciduous forests, is very complex and variable, thus the traditional grid model and geometric model cannot construct detailed features, and the original TIN model cannot eliminate the effects of LiDAR points under the canopy surface. To reconstruct the forest canopy with more detail and high accuracy, a local scene-based TIN was generated in this paper. With the new method, the neighboring sets of each LiDAR point were first defined and identified using Delaunay triangulations. Then, control points were extracted by analyzing multiple local scenes, which include the shapes of steeple, inverted tip, ridge, saddle, and horseshoe. Finally, the scene-based TIN was generated using control points as nodes.

## 2. Material and Methods

### 2.1. Study Area

The case study was located in the neighborhood of Upper East Side, Milwaukee, Wisconsin, the United States (Figure 1). This is a residential neighborhood with four blocks between Lake Michigan to the east and the Milwaukee Rive to the west. This area is not only covered by relatively flat roads and lawns, but also by tall residential buildings, various deciduous and conifers trees, and shrubs, etc. Most buildings have smooth but sloping roofs while the canopy surfaces are irregular and rough. Crown sizes, tree heights, and crown shapes vary greatly, with the radius of the tree crown in the study area ranging from 2 to 15 m and tree heights from 3 to 25 m. Deciduous trees, such as ash (*Fraxinus* spp.), maples (*Acer* spp.), oak (*Quercus* spp.), and others, have large crowns, but the crown sizes of the coniferous trees, including pine (*Pinus* spp.) and spruce (*Picea* spp.), are relatively small.

### 2.2. Data Set

LiDAR data and hyperspectral images with the airborne imaging spectrometer for application (AISA) were collected in September 2010 and August 2008 by Terra Remote Sensing Inc. (TRSI), and then they were preprocessed by Native Communities Development Corporation Imaging (NCDC) and SRA International Inc. (SRC), respectively. The provided LiDAR data has a mean density of 1.6 points per m$^2$, with the coordinate system of NAD27 State Plane Wisconsin South (FIPS 4803). The AISA image has a spatial resolution of 1.0 m with a spectral resolution of 4.6 nm in the visible and near infrared wavelengths and 6.26 nm in the short-wave infrared wavelength. AISA images were used as references to visually test LiDAR based 3D models.

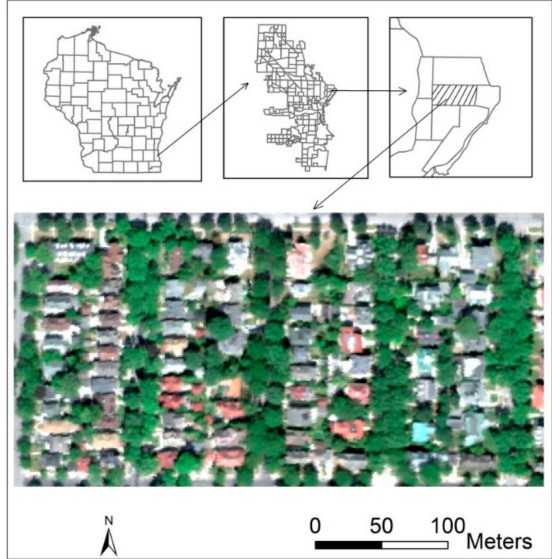

**Figure 1.** Study area (neighborhood of the Upper East Side of Milwaukee, Wisconsin, United States).

*2.3. Methods*

The quality of a TIN model depends on control points, as effective points can produce accurate models while erroneous points tend to produce deformed ones. To improve the accuracy of 3D canopy models, it necessary to analyze the morphological characteristics of canopies in detail and extract the effective control points with the shape of steeple, ridges, saddles, and crown edges. As the identification of these points depends on the spatial relationship with their neighbors, the model can be constructed with three steps: (1) determining the set of neighbors of each point using Delaunay triangulation, (2) extracting control points based on local scenes, and (3) creating the canopy model with the TIN algorithm.

2.3.1. Neighbor Identification

The irregular distribution of LiDAR points indicates that their spatial proximity cannot be analyzed using regular grids. Points that are closer in space but located on different planes or slopes cannot be considered as neighbors on the surface, while points that are relatively far apart may be neighbors due to the lack of intermediate points on the same plane. In order to analyze the spatial relationship of point sets, the proximity relationship is first determined by Delaunay triangulations. A triangulation is a collection of triangles with empty circumcircles, which contain the edges' endpoints but not any other points. The neighbors of each point are defined as the points that connect to it in the local irregular Delaunay triangle network.

For points in a two-dimension plane, the fundamental criterion is the empty circumcircle. Any Delaunay triangulation ensures the circumcircle of a triangle contains no other points in its interior. The Delaunay and non-Delaunay triangulations for four points were compared in the following illustration (Figure 2). Connecting the four points, two irregular triangles can be created. If the circumcircle associated with T1 is empty, indicating no other points are in the circumcircle, this triangulation is a Delaunay triangulation. However, if the circumcircle associated with T1 contains V3 in its interior, this triangulation is not a Delaunay triangulation.

With Delaunay criterion, the triangle irregular network was generated based on the 3D coordinates of LiDAR points. The points connected by a line segment are considered to be adjacent, and all the points connected to a vertex are defined as the neighbors of this point. As shown in Delaunay triangulation, both v2 and v3 are the neighboring points of v1, even though they have different distances to v1.

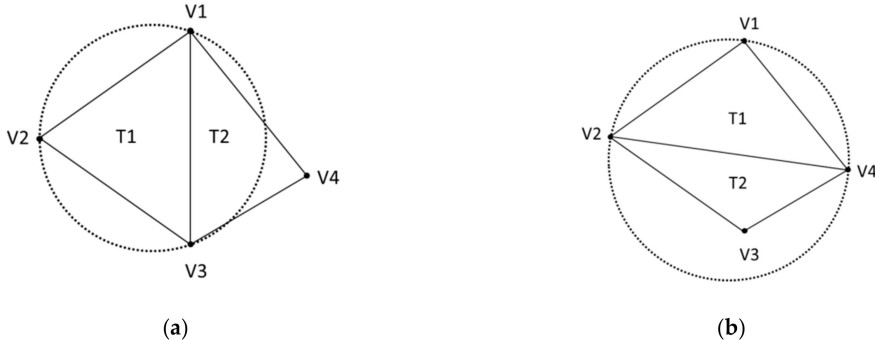

**Figure 2.** Comparison of Delaunay and non-Delaunay triangulations. (**a**) Delaunay triangulation; (**b**) non-Delaunay triangulation.

### 2.3.2. Scene-Based Spatial Relationship

The crown morphology is complex and variable, so the interpolated raster model, the convex polygon model and the geometric model cannot express detailed crown features due to the loss of many real point data. However, the precise selection of control points helps to reveal the detailed properties and overall shape profile of the canopy because the key points are retained to the greatest extent. To accurately extract these control points, morphological features of the canopy were analyzed based on local scenes: steeple shape, inverted tip shape, ridge shape, saddle shape, and horseshoe (Figure 3).

(1) Steeple shape: the center point is located at the highest position of the neighborhood and no other point is higher than the center. The center point is considered to be the point on the surface of the canopy and remained to produce the TIN model.

(2) Inverted tip shape: the center point is the lowest one within the local triangle network, which is assumed to be the point that penetrates the canopy surface and is deleted in the process of TIN model generation.

(3) Ridge shape: the points on either side of the ridge are lower than the points on the ridge line. Although the point at the ridge line does not necessarily have the highest value, it is considered to be on the surface of the branch and remained for canopy model generation.

(4) Saddle shape: the points on both sides of the saddle are higher than the points in the middle of the saddle. If the center point of the saddle is greater than the value of either end of the saddle, the point is considered to be the connection point of the branches and remains as a key point to create the TIN model. Otherwise, the point is removed as a point penetrating the surface.

The horseshoe shape is a special case of the saddle shape. Some points around the center are higher than the center point, but the other points are lower than the center point. Since the center point is neither the highest nor the lowest one, neither at the top of the ridge nor at the bottom of the saddle, the value is checked by the interpolation method to determine if it needs to be used in the model creation. With the IDW method, the discriminant value was calculated using neighboring point values and their distances as parameters, and then compared with the center value.

(5) If the point's value is higher than the discriminant value, it remains as the point on the surface and used to create the TIN model.

(6) If the point's value is lower than the discriminant value, it is removed as a breakthrough point, and only the remaining points are used to create the TIN model.

In the process of control point extraction, the recursive algorithm is also applied. When a point is deleted as an outlier, the proximity of the points will change and the new scene will be presented, thus the remaining points need to be continually analyzed until all points in the study area meet the criteria.

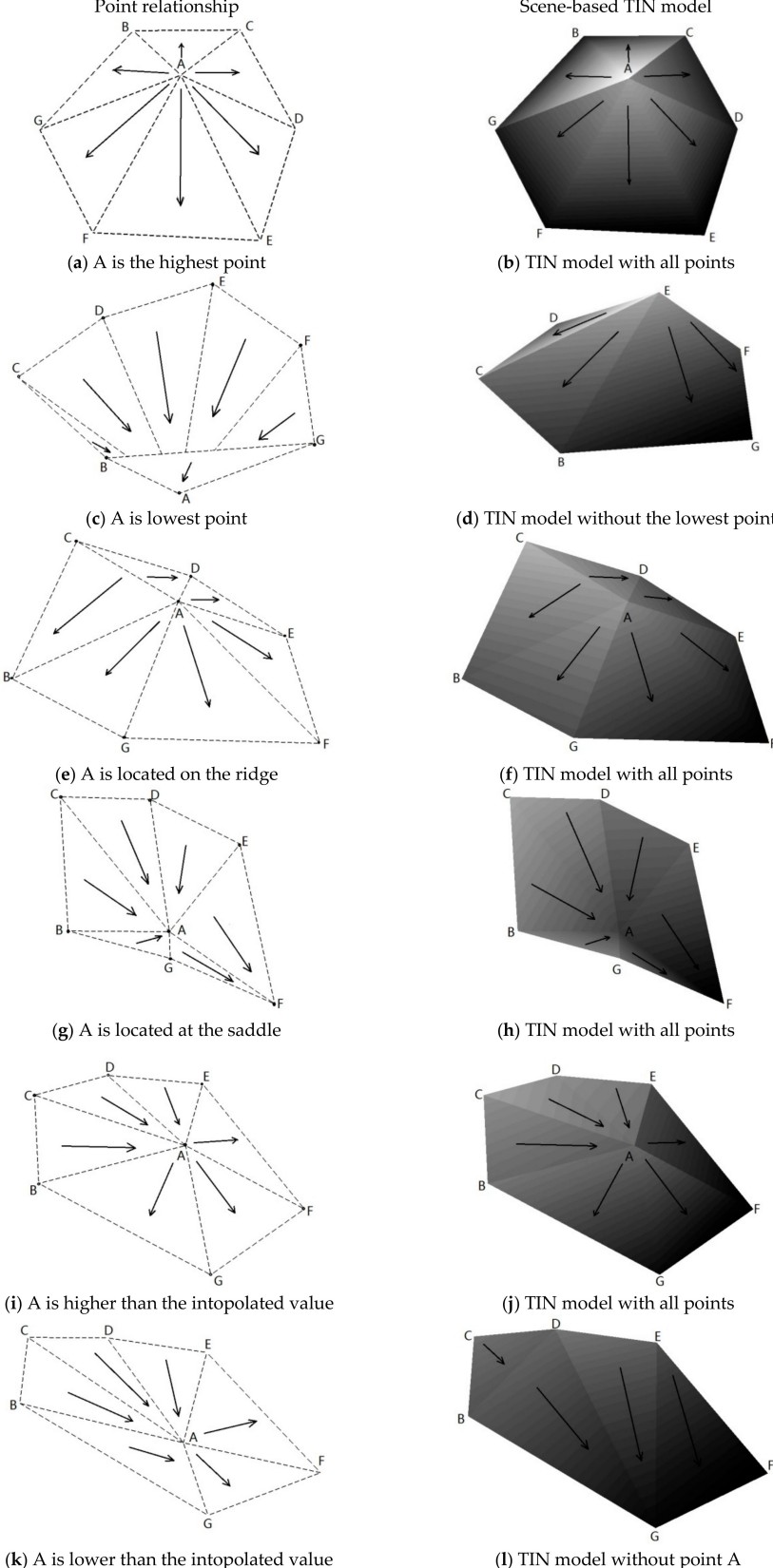

**Figure 3.** Five situations of spatial relationship. (**a**) Steeple shape, (**b**) Steeple shaped TIN model, (**c**) Inverted shape, (**d**) Inverted shaped TIN model (**e**) Ridge shape, (**f**) Ridge shaped TIN model (**g**) Saddle shape, (**h**) Saddle shaped TIN model, (**i**) Horseshoe shape 1-point higher than the discriminated value, (**j**) Horseshoe shaped TIN model 1, (**k**) Horseshoe shape 2-point lower than the discriminated value, (**l**) Horseshoe shaped TIN model 2.

### 2.3.3. Canopy Model Creation

A canopy model was created using a triangular irregular network (TIN) with the control points as nodes. These points are connected by lines to form a network of contiguous but non-overlapping triangles to represent the crown surfaces, which satisfies the requirement that a circle drawn through the three nodes of a triangle will contain no other node.

### 2.3.4. Accuracy Assessment and Comparative Analysis

The scene-based CHM is first visually compared with the trees in the Google earth maps and AISA hyperspectral images, and then the height errors of all the single trees along the roads were calculated and analyzed. Since the control points are the nodes of the TIN, there is no error between the TIN and the LiDAR at the control points, but the difference between LiDAR and TIN-derived raster images can be used to represent the height error of an entire tree crown, considering the interaction of spatial patterns on the model. With regards to this, the scene-based TIN model was converted to the raster image, and the average and standard deviation of the height errors were calculated and compared with other models.

In order to observe the local performance of the model in detail, eight randomly selected individual trees, including two ash trees, two maple trees, two oak trees, and two pine trees, were examined (Figure 4). Ash, maple, and oak are deciduous trees with large crowns, but pine trees are coniferous trees with small crowns.

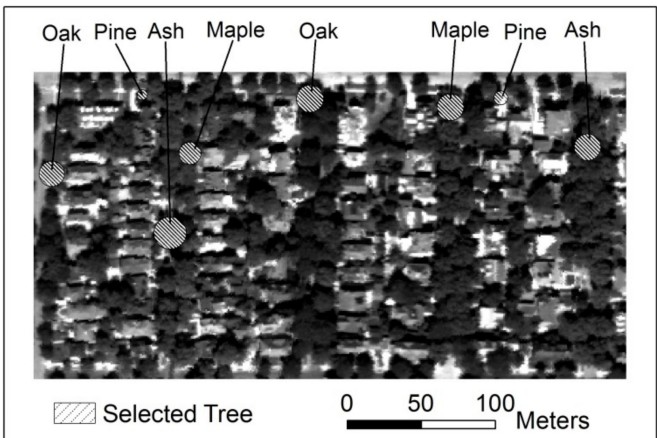

**Figure 4.** Selected trees.

In addition, a quantitative analysis was performed to examine the accuracy of the developed model, and a comparative analysis was conducted with three traditional methods, namely the original raster, the smooth raster, and the original TIN-derived (ori-TIN-derived) raster. For such a comparison, the scene-based TIN model was converted into a grid raster with a spatial resolution of 0.2 m. Since each value was determined by the surrounding points, the errors of the raster image at control points were calculated to represent the model accuracy at the entire crown level. To perform comparison, the original raster image was created using an IDW interpolation method with original points; the smooth raster image was generated by filtering the original raster with a low-pass filtering method; and the ori-TIN-derived raster was obtained by converting the Original LiDAR-derived TIN model to a grid image, with the same cell size as the developed model. In the study area, 155 trees along the road were selected and manually extracted for error analysis. The absolute difference between each LiDAR point and the corresponding grid value was calculated as height error of the raster at the LiDAR point position.

## 3. Results

### 3.1. Control Point Extraction and 3D Model Generation

With the local Delaunay triangulations, two points connected by a line segment are defined as neighbors, and all points connected to a target point are neighboring point sets of the point. Within the set of neighbors, multiple scenarios were analyzed and the control points were extracted by a recursive algorithm, and their positional relationship in the 2D plane is shown in Figure 5.

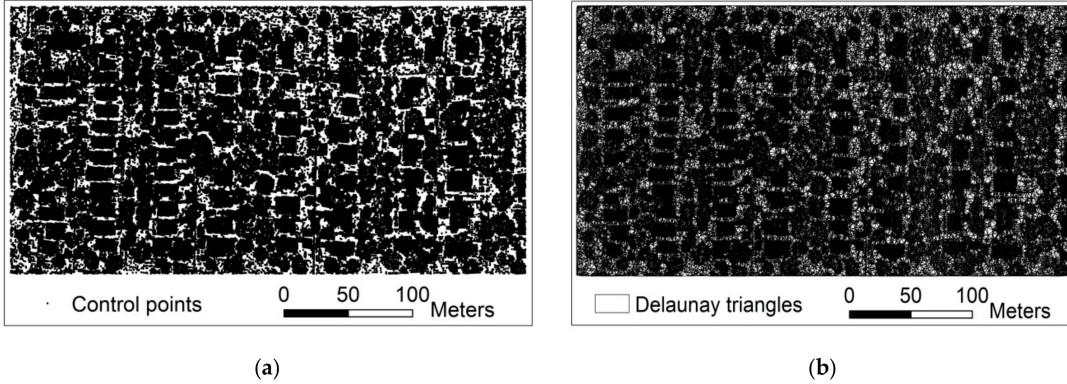

(**a**)　　　　　　　　　　　　　　　　　　　(**b**)

**Figure 5.** Control point distribution and Delaunay triangles of the 2D plane. (**a**) Control point distribution; (**b**) Delaunay triangles.

Since a lot of the redundant and abnormal points were removed, the density of points was greatly reduced. The number of control points was only 58.47% of the original point clouds. From the control point distribution map, it can be found that the point density varies with the objects. Buildings have the most control points, reaching 96.85% of the original density. The second forest canopy has the density of 72.41% of the original points, and the flat ground or lawns has the sparsest control points, only 20% of the original ones.

With the control points, a 3D triangular irregular network was created (Figure 6). By visual observation, we can find the scene-based model simulates the shape of the canopy very well. First, the apex of the tree and the ridge of the branch were accurately constructed. Since the highest points within the neighboring area were retained and directly served as the nodes of the TIN model, there is no error in the heights of the trees in comparison with the original point cloud. Moreover, the points on the ridges were extracted for the TIN model, so the skeleton of the branches and the overall outline of the canopy were displayed. Second, valley lines and the boundaries of canopy were clear. The points located in the saddles were identified and retained as the branch lines, so the undulating shape of the canopy could be revealed. In contrast, the traditional methods, including low-pass filtering, top DSM, sorting and selecting mechanism, always deleted these points due to their low values, and therefore ignored the detail features of the crown. Third, the modeled surface was smooth. As the abnormal points appearing in the saddle, horseshoe, and inverted tip scenes were removed, and the points that penetrated the canopy surface no longer affected the canopy height model, there were no obvious pits on the surface.

### 3.2. Accuracy Assessment and Comparative Analysis

A comparison between the scene-based TIN model and the AISA image reveals that the morphological features of the newly created objects are consistent with those in the hyperspectral image (Figure 7). The scene-based TIN model not only provides the overall shape of the objects, but also describes the details. The buildings have smooth roofs, the trees have relatively clear boundaries, and the crowns have varying branches over the new model. As the local scene-based morphological

analysis contributes to this model, the retained LiDAR points can describe the shapes to the maximal extent and eliminates the penetration and abnormal points, thus represent the tree crown well.

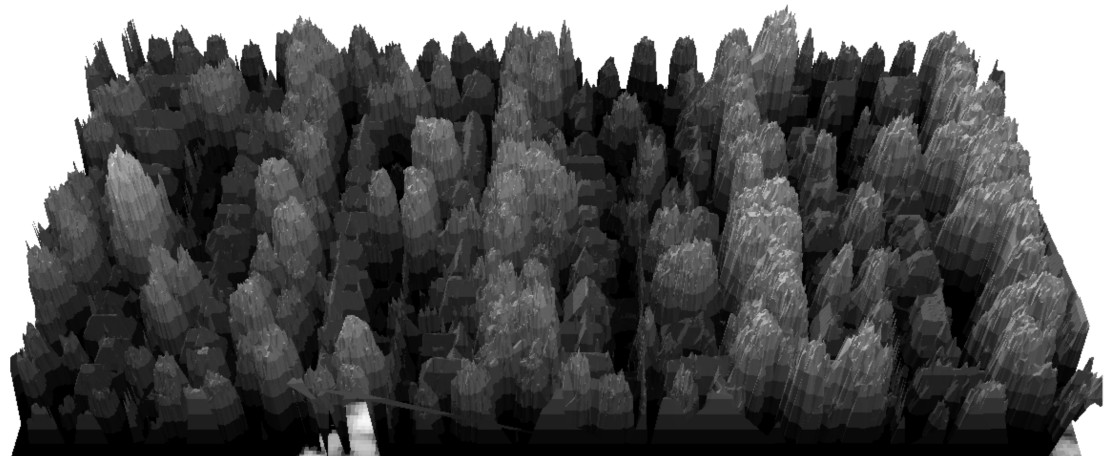

**Figure 6.** Scene-based triangular irregular network (TIN) model.

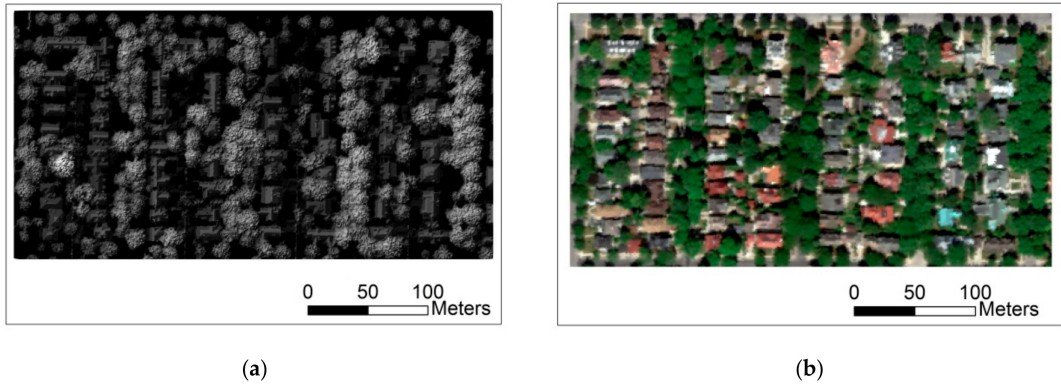

(**a**)                                                                 (**b**)

**Figure 7.** Comparison between the scene-based model and airborne imaging spectrometer for application (AISA) image. (**a**) Scene-based model; (**b**) AISA image.

For the extracted 155 trees in the study area, statistical results indicate that the average error of the scene-based model is 0.18 m, with a standard deviation of 0.41, while the errors caused by the original raster, smoothed raster, and ori-TIN derived raster are 0.96, 2.05, and 1.00 m, with standard deviations of 1.61, 1.82, and 1.87, respectively (Table 1). With the scene-base model, the error mean is reduced by 81.25%, 91.22%, and 82% from those generated by the above three traditional models, and the fluctuation range of the errors has also become very narrow, which is only 18%, 10%, and 19% of error ranges caused by ori-raster, smooth raster, and ori-TIN derived raster (Figure 8). For the scene-based model, 69.03% of the trees are with an error less than 0.2 m, 27.74% are less than 0.3 m, and only 3.23% are between 0.3 and 0.51 m. In contrast, 99.35% of trees in original raster have the errors greater than 0.2 m, of which, 66.45% of trees are with errors ranging from 0.2 to 1.0 m, and 33.55% are greater than 1.0 m. The absolute errors from the original TIN model are from 0.1 to 2.2 m, which is similar to the error distribution of original raster. Although the low-pass filtering algorithm smooths the surfaces of the model, it produces relatively large errors. Especially, the percentage of points with errors greater than 1.0, 2.0, 3.0, and 4.0 m reach 13.55%, 37.42%, 43.87%, 5.16%, respectively. The error mean and its distribution indicate that the scene-based TIN is relatively stable in modeling forest canopies and is less affected by tree species or canopy morphologies.

**Table 1.** Comparison of absolute error mean and standard deviation.

| Models | Absolute Error Mean (m) | Standard Deviation |
|---|---|---|
| Original raster | 0.96 | 1.61 |
| Smoothed raster | 2.05 | 1.82 |
| Ori-TIN derived raster | 1.00 | 1.87 |
| Scene-based TIN derived Raster | 0.18 | 0.41 |

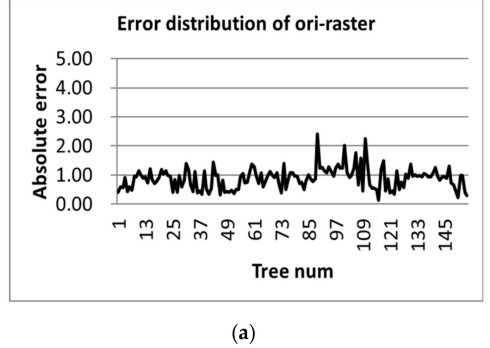

(**a**)

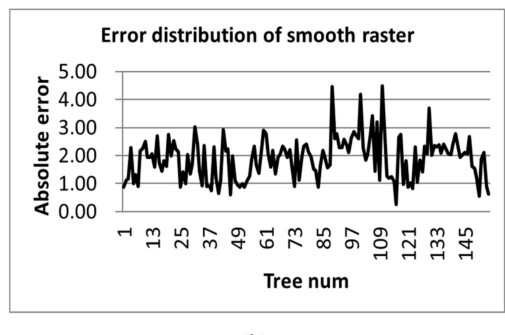

(**b**)

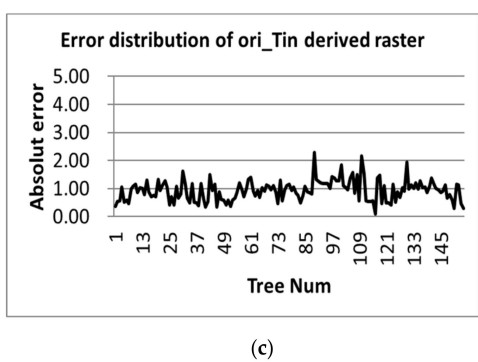

(**c**)

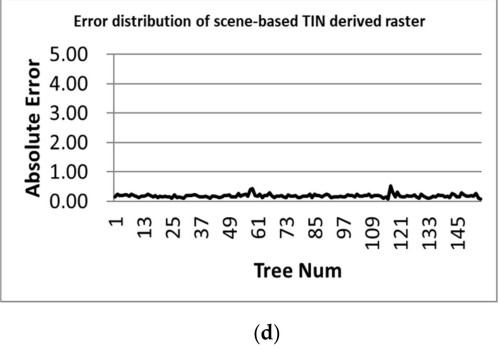

(**d**)

**Figure 8.** Error distribution of four different models. (**a**) Error distribution of ori-raster; (**b**) error distribution of smooth raster; (**c**) error distribution of ori_TIN derived raster; (**d**) error distribution of scene-based TIN derived raster.

To observe and analyze the new model in the details, eight individual trees were extracted from the scene-based TIN model and compared with the original TIN models in 2.5D and 3D perspectives (Figure 9). The scene-based models have smooth surfaces with good crown morphologies, especially protrusions, depressions, grooves, and boundaries of the crown are obvious. Comparatively, the original TIN models exhibit a number of pits, similar to the raster image obtained by the interpolation method. In particular, the deciduous trees, such as ash, maple, and oak trees have relatively serious pit phenomena, largely because the relatively low closure of the deciduous forest canopy leads to a higher penetration of the laser, resulting in a rough model surface. The new method is to identify and eliminate these points by analyzing morphologies of the local scenes, thus the model created with only control points is more continuous and smoother. Whereas, conifers, such as pines, have small but dense crowns, so fewer points can penetrate the canopy and the resulting original TIN models are relatively smooth, and the new method plays a limited role in improving coniferous models.

Scene-based TIN and ori-TIN models of the eight trees were also converted into grid images, and compared with original raster and low-pass filtered raster in 2D perspective (Figure 10). Similar to 2.5D and 3D images, large deciduous crowns, including ash, maple, and oak trees exhibit very distinct pit phenomena over both original and ori-TIN derived raster images, while small coniferous canopies, especially Pine 1, present relatively few pits. Considering different tree species have different canopy

porosity, which in turn leads to differences in model smoothness, it can be said that these models are sensitive to tissue distribution and tree species. In contrast, the scene-based TIN model produced smooth but clearly textured canopy model for all kinds of trees, and therefore the new model has better applicability for trees with different structure forms.

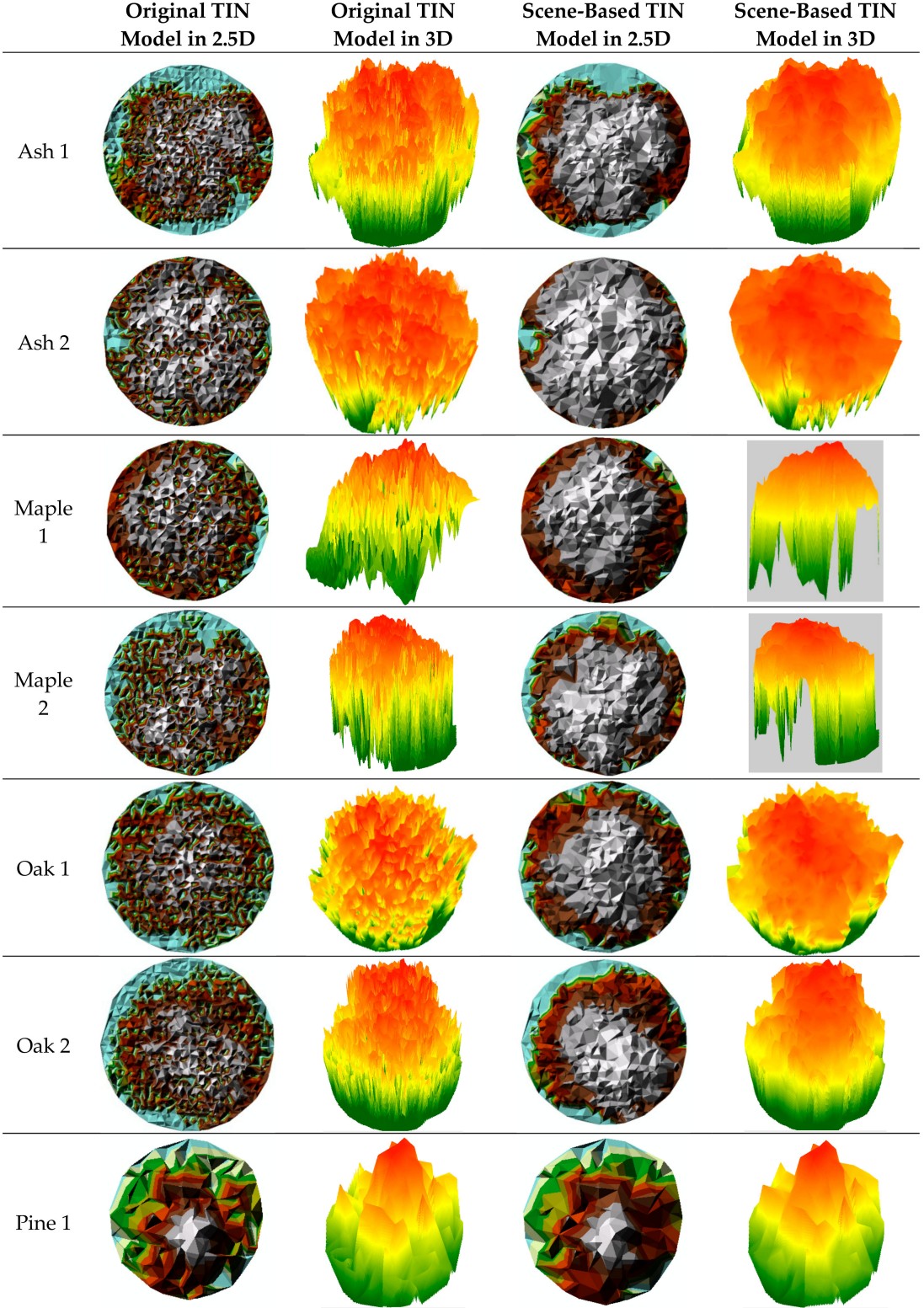

**Figure 9.** *Cont.*

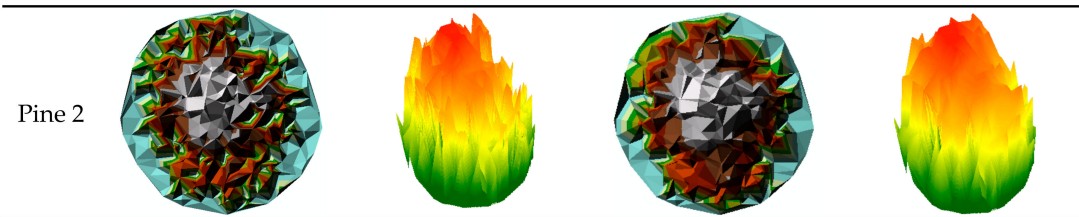

**Figure 9.** 2D and 3D vision of eight individual trees.

The average errors and standard deviations of the eight different trees were also calculated from the rasterized images and LiDAR points (Table 2). With Original Raster images, the lowest error is 0.23 m for Pine 1 and the highest error is 1.49 m for Maple 1. Ori-TIN derived raster resulted in similar errors, ranging from 0.29 to 1.53 m, with a standard deviation from 0.43 to 2.33. Different from low-pass filtered raster which increased the average error of Pine 1 to 0.55 m and Maple 1 to 3.11 m, Scene-based-TIN-derived raster images reduced the average error of eight trees to 0.18 m, with an average standard deviation of 0.37 (Figure 11). Especially, the average estimation error for the six deciduous trees is 0.15 m with a standard deviation of 0.37, while the error mean for the two pines is 0.25 with a standard deviation of 0.35.

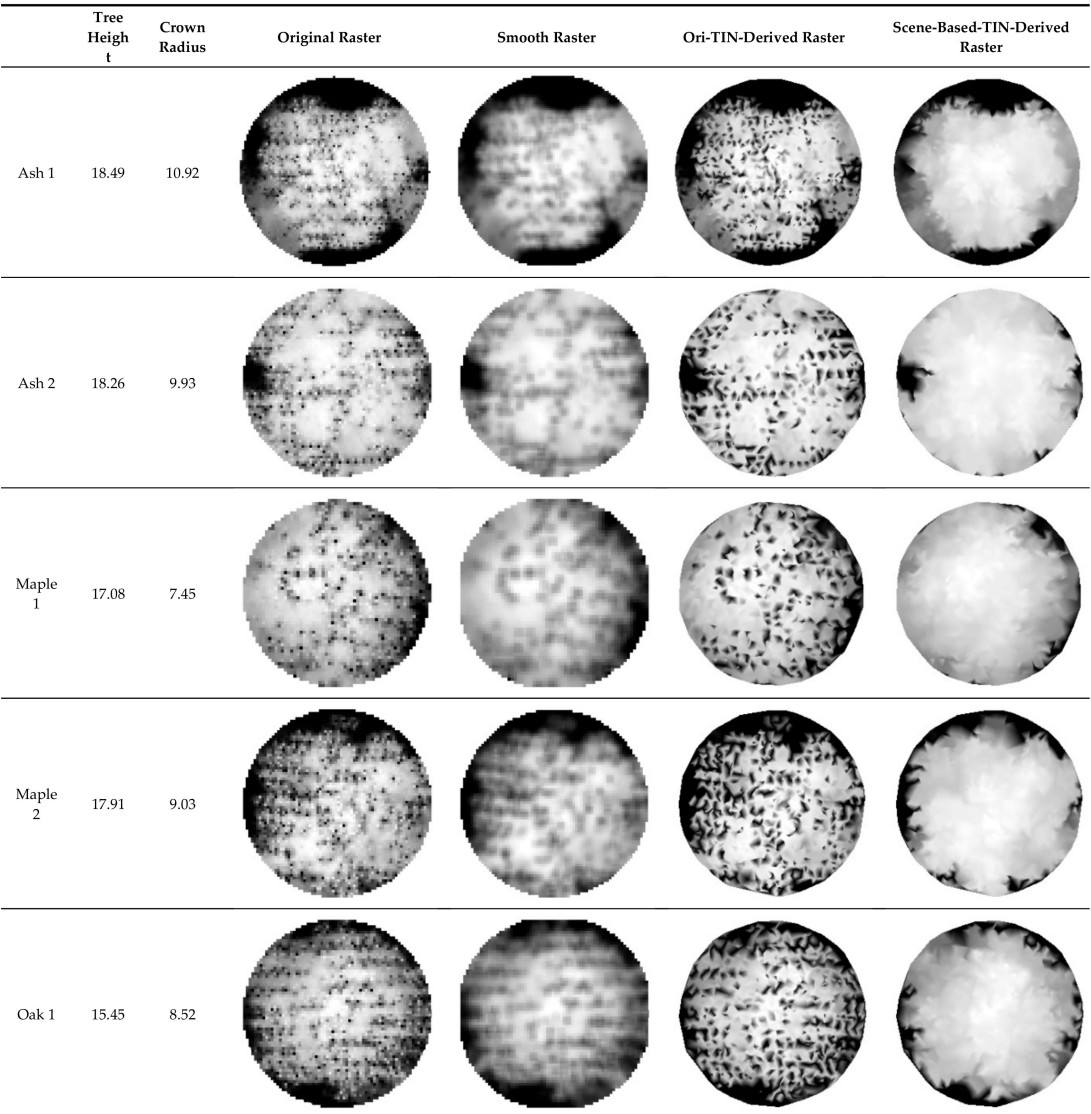

**Figure 10.** *Cont.*

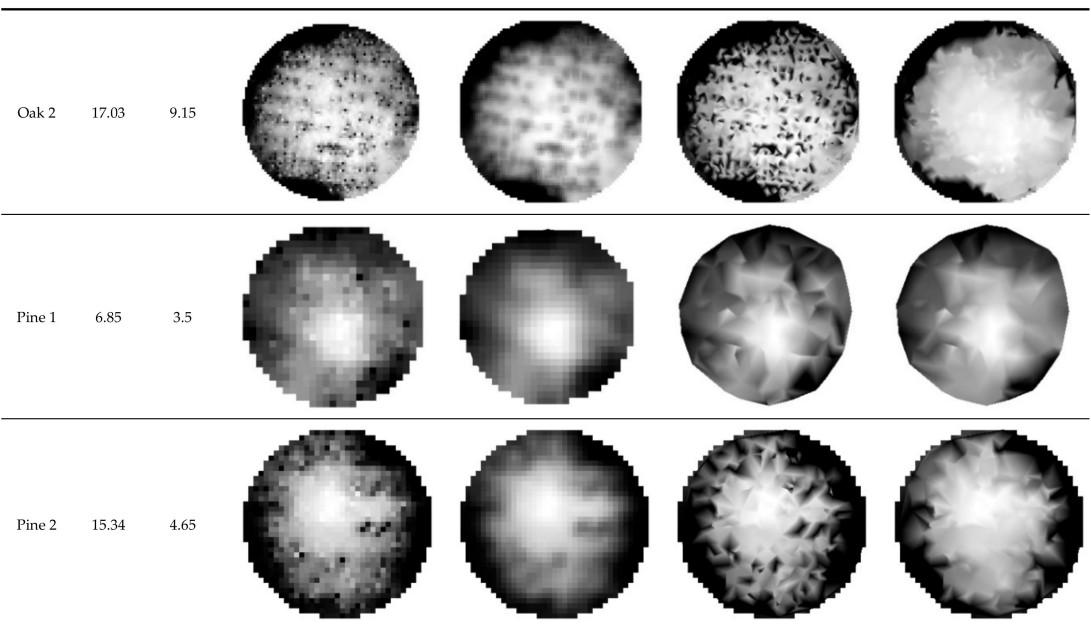

**Figure 10.** Raster image comparisons.

**Table 2.** Raster errors at control point positions.

| Trees | Average of Errors | | | | Standard Deviation | | | |
|-------|-------------------|---|---|---|--------------------|---|---|---|
| | Original Raster | Smooth Raster | Ori-TIN-Derived Raster | Scene-Based-TIN-Derived Raster | Original Raster | Smooth Raster | Ori-TIN-Derived Raster | Scene-Based-TIN-Derived Raster |
| Ash 1 | 1.17 | 2.42 | 1.16 | 0.20 | 1.76 | 1.79 | 2.03 | 0.47 |
| Ash 2 | 0.90 | 2.07 | 0.91 | 0.11 | 1.69 | 1.77 | 1.81 | 0.22 |
| Maple 1 | 0.99 | 2.05 | 1.08 | 0.16 | 1.53 | 1.65 | 1.97 | 0.28 |
| Maple 2 | 1.49 | 3.11 | 1.53 | 0.20 | 2.08 | 2.03 | 2.33 | 0.41 |
| Oak 1 | 1.14 | 2.50 | 1.15 | 0.117 | 1.59 | 1.62 | 1.65 | 0.31 |
| Oak 2 | 1.06 | 2.34 | 1.20 | 0.126 | 1.43 | 1.65 | 1.91 | 0.53 |
| Pine1 | 0.23 | 0.55 | 0.29 | 0.19 | 0.36 | 0.45 | 0.43 | 0.30 |
| Pine 2 | 0.87 | 1.82 | 1.10 | 0.31 | 1.34 | 1.63 | 1.83 | 0.40 |

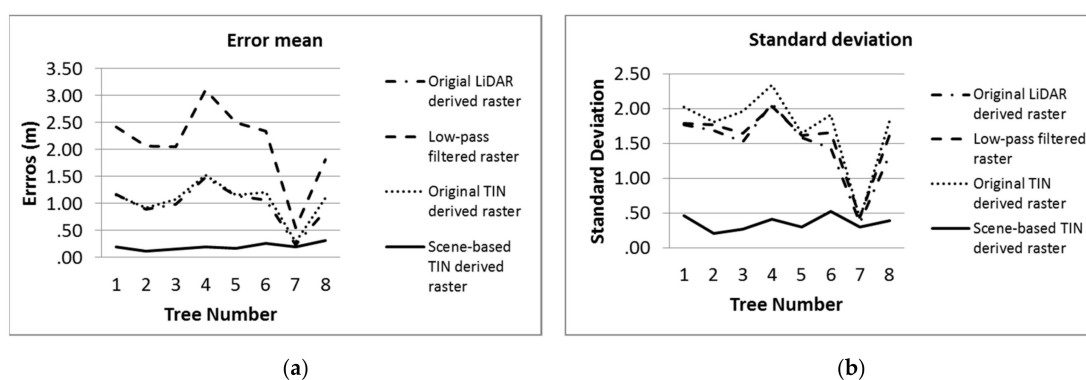

**Figure 11.** Average errors (**a**) and standard deviation (**b**).

## 4. Discussion

Scene-based TIN is the best technique for forest canopy modeling in comparison with the CHM model, original TIN model, and geometric model. This new model has the ability to provide accurate canopy morphologies and height information, while CHM and original TIN models always underestimate the canopy heights and change crown shapes due to the influence of point penetration [39,40]. The scene-based model is also suitable for mixed forests without any preassumption,

but the creation of the geometric model is usually based on specific crown shapes (e.g., convex, cone, hemisphere, and half-ellipsoid) [41,42], so it has limitations in forests with different crown shapes

3D reconstruction of the tree canopy is an important means for forest morphology illustration, individual tree segmentation, and biophysical parameter extraction [43]. However, accurate reconstruction of real trees from LiDAR points is a huge challenge [44]. On the one hand, tree crowns are highly complex in geometry and topology. In particular, the overall shape of a single crown varies with different tree species, and the geometric shape of different portions of the same crown are also different. Moreover, a real crown may have a shape of concavity [45]. On the other hand, LiDAR points have no topological connections; these points are unstructured and unevenly distributed, and are sampled from discrete tissue elements [46], thereby difficult to identify surface points and define typical boundary points in a tree crown.

Classical surface reconstruction techniques fail to obtain 3D models of vegetation, although they perform well in creating nonvegetative models [47]. Interpolation is a common method to predict unknown values for any geographic point data based on the assumption that spatially distributed objects are spatially correlated. However, the resulting CHM contains many pits because laser beams penetrate deeply into a tree crown, which negatively affect tree detection and subsequent biophysical measurements [48]. With the IDW interpolation, the average height of 155 trees in this study was underestimated by 0.96 m. To fill the invalid values in LiDAR-derived CHMs, low-pass filter was applied in this study. Although the algorithm has the ability to smooth canopy height models, it also reduced the highest point value and increased the lowest point value, therefore the average error of 155 trees was increased to 2.05 m.

Vector-based TIN model is an alternative to the grid-based representation of forest canopy, which can avoid errors brought by mesh sizes or interpolation methods [49]. In a TIN, all points that make up the model are the triangle nodes, which not only preserve the highest values that represent the height of a tree, but also contain the outliers that penetrate the surface. Therefore, the surface of the model is still rough and the average height error is not reduced, the resulting original TIN error (1.0 m) in this study is very similar to CHM error (0.96 m). As the generation of CHM and TIN models is based on a common spatial correlation principle, the approximate shape of the canopy can be revealed. However, these models do not take the specific characteristics of vegetation, such as randomly distributed gaps, rich concavity, and undulating branches, into account. Therefore, these models ignore the local form of the tree crown and produce large errors.

The scene-based TIN model overcomes the above problems by analyzing five morphological scenarios, including steeple, inverted tip, ridge, saddle, and horseshoe shapes. As a result, local maxima are retained as vertexes of branches or treetops, but local minima are removed as the outliers. Although points on ridges are not the highest ones, they are still regarded as important controls due to their locations on the convex surface. The points sitting at saddles are generally lower than most surrounding points, but if they can form recessed boundary lines with other points instead of being isolated, they are reserved as edge points of crowns or local branches. Additionally, points located inside horseshoes are tested using traditional interpolation methods to determine whether to keep or reject them. Finally, key points are extracted, and outliers are eliminated through scene analysis.

The scene-based TIN model produces a smooth canopy surface. The overall outline of the canopy is consistent with the AISA hyperspectral image, and the detailed information of individual crowns, such as branch distribution and branch boundaries, are well depicted. Statistical analysis indicates the new model reduces overall height error to 0.18 m, which is only 18.75% of the original CHM, 8.78% of low-pass CHM, and 18.00% of original TIN. The low error of the new model is mainly attributed to the removal of outliners and the use of critical control points, including points at the highest position, ridgeline, and saddle line.

The scene-based TIN model is suitable for both deciduous and coniferous crowns. The structural differences among forests lead to different porosity of the canopies, which in turn affects the penetration of the laser. Therefore, the laser has strong penetration ability to a relatively sparse canopy and weak

penetrating ability to a dense canopy. Due to the neglect of the structural characteristics of the canopy, the original interpolation method, the low-pass filtering, and the original TIN algorithm produce a higher error for deciduous trees. However, the scene-based TIN model analyzes various possible local morphologies and extracts the control points to depict the structural features. Without outliers, both deciduous and coniferous forest are accurately modeled. In comparison with the original, smooth, and ori-TIN derived raster data, the errors were reduced by 86.0%, 93.0%, and 87.0% for deciduous trees, and 54.0%, 78.0%, and 63.0% for coniferous trees, respectively.

In addition to the advantages of producing a better model, the extraction of control points reduces the amount of data storage and differentiates the point density for target objects. Since some points covering trees were identified as penetrating points and deleted, only the points over the canopy surface were preserved as control points. Different from the penetration of LiDAR into trees, laser cannot penetrate building roofs, and thus most points, covering buildings, were preserved to model their gentle slopes. Compared to trees and sloping roofs, the surfaces of grass and the road are relatively flat, and a large number of points were deleted as local depressions and few points were defined as control points to create their flat surface models. The result is consistent with common sense that more points are needed in rugged areas and fewer points in relatively flat areas [31].

## 5. Conclusions

In this article, a scene-based TIN algorithm was developed to create an accurate canopy height model. With the new method, the irregular local triangles were first formed based on the Delaunay triangulation principle, then control points were extracted by analyzing local scenes, and finally the scene-based TIN model was generated using control points as nodes.

The core idea of the new method is to utilize local scenes to determine the spatial position of laser points for control point extraction and model generation, especially steeple, inverted tip, ridge, saddle, and horseshoe shapes are defined as the basic morphological unit of canopies. Points on convex surfaces, such as steeples and ridges, are considered as height control points to represent the surface height. Points on saddle lines are viewed as boundary control points to divide different region shapes. Points located at inverted tip are determined as outliers penetrating the crown surface and eliminated. Additionally, other points within horseshoe shapes are tested by traditional interpolation method to decide whether to keep them.

The extraction of control points greatly reduces the storage space of LiDAR data. In the entire study area, 41.53% of point data were deleted, which is especially meaningful in the era of massive data. In particular, 96.85% of points on buildings, 72.41% of points over trees, and 23.28% points on ground or lawns were retained. Such a significant difference of point density between tree canopy and the ground provides a possibility for the further identification and segmentation of tree canopies.

The scene-based control point extraction and scene-based TIN algorithm are suitable for forest height models with various environments. The similar accuracy of deciduous and coniferous models indicate that scene-based TIN can be used in mixed forests, regardless of tree species. However, this method was only applied in LiDAR data with a specific spatial resolution, so more experiments with different point resolutions need to be performed in the future.

**Author Contributions:** Conceptualization, H.L.; Formal analysis, H.L.; Funding acquisition, H.L.; Investigation, H.L.; Methodology, H.L.; Project administration, C.W.; Resources, C.W.; Software, H.L.; Supervision, C.W.; Validation, C.W.; Visualization, C.W.; Writing—original draft, H.L.; Writing—review and editing, C.W. All authors have read and agreed to the published version of the manuscript.

**Funding:** This research was funded by the Natural Science Foundation of Zhejiang Province (grant number LY19D010005).

**Conflicts of Interest:** The authors declare no conflict of interest.

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
