# Peer review of "Developing a Scene-Based Triangulated Irregular Network (TIN) Technique for Individual Tree Crown Reconstruction with LiDAR Data"

_forests, doi:10.3390/f11010028_

Round 1

Reviewer 1 Report

Manuscript ID: forests-660364 entitled: Developing a scene-based triangulated irregular network (TIN)

technique for individual tree crown reconstruction with LiDAR data.

General comments:

This manuscript presents a very important problem related to the LiDAR-based individual tree crown reconstruction. The Authors aim to find the more efficient method of reconstruct canopy morphologies then previously used, such as: LiDAR-derived rasterization, low-pass filtering smooth algorithm, and original TIN model, which are difficulties in balancing morphological accuracy and model smoothness. Proposed by Authors scene-based triangulated irregular network (TIN) was generated with three steps based on the local scene principle: i) triangulation, ii) determining key control points and iii) using them as nodes to developed the TIN model. This is a very useful study which allows to accurately reconstruct different canopy shapes with a relatively smooth surface. Statistical analysis carried out by the Authors confirmed that the overall error of the new model is smaller from 5 to 10 times than others three compared methods.

Despite the fact that this is the work on a very important issue well suited to the profile of the forests journal, the Authors did not avoid shortcomings.

The quality and relevance of the datasets appears to be high. However, the biggest problem is the weakness of a real discussion in this manuscript. The main concern is that Authors findings does not have a proper discussion. I would strongly recommend adding it. In the introduction, the Authors correctly refer to the results of other researchers. In relation to this fact there are many references to previous research made by the international researchers, so no difficult to access for the publications. On the other hand, there are some relevant references for this paper that could be included and can be of help for the discussion and also for giving explain better the results obtained.

Specific comments:

Results & Discussion

The results section isn’t clear, and the discussion section isn't here at all. The reader has the impression that there is no discussion here. There are no references to the results of other authors. This is unacceptable. The Authors should separate the results and add the discussion. Throughout the discussion the Authors should reference to the results of similar studies, particularly those which make use of three mentioned traditional methods. There is a decent amount of literature on this, which authors largely fail to acknowledge throughout the manuscript. I am left with no clue as to how your results compare to other results. This is a serious shortcoming of the manuscript.

Citing references in the text

The Authors should fit the manuscript text and references lists to MDPI’s citations style based on the style used by the American Chemical Society (ACS). Numbers [1] instead name and year (Tasoulas et al., 2013). This needs to be corrected throughout the text.

Summarizing review, this paper is unsuitable for publication in its current form. Authors should rewrite some parts of the work. Once these changes are made, this paper is a reasonable contribution to Forests Journal.

Author Response

Response to Reviewer 1 Comments

General comments:

This manuscript presents a very important problem related to the LiDAR-based individual tree crown reconstruction. The Authors aim to find the more efficient method of reconstruct canopy morphologies then previously used, such as: LiDAR-derived rasterization, low-pass filtering smooth algorithm, and original TIN model, which are difficulties in balancing morphological accuracy and model smoothness. Proposed by Authors scene-based triangulated irregular network (TIN) was generated with three steps based on the local scene principle: i) triangulation, ii) determining key control points and iii) using them as nodes to developed the TIN model. This is a very useful study which allows to accurately reconstruct different canopy shapes with a relatively smooth surface. Statistical analysis carried out by the Authors confirmed that the overall error of the new model is smaller from 5 to 10 times than others three compared methods.

Despite the fact that this is the work on a very important issue well suited to the profile of the forests journal, the Authors did not avoid shortcomings.

The quality and relevance of the datasets appears to be high. However, the biggest problem is the weakness of a real discussion in this manuscript. The main concern is that Authors findings does not have a proper discussion. I would strongly recommend adding it. In the introduction, the Authors correctly refer to the results of other researchers. In relation to this fact there are many references to previous research made by the international researchers, so no difficult to access for the publications. On the other hand, there are some relevant references for this paper that could be included and can be of help for the discussion and also for giving explain better the results obtained.

Response 1:

Discussion section has been added in the article and more references were included.

Specific comments:

Results & Discussion

The results section isn’t clear, and the discussion section isn't here at all. The reader has the impression that there is no discussion here. There are no references to the results of other authors. This is unacceptable. The Authors should separate the results and add the discussion. Throughout the discussion the Authors should reference to the results of similar studies, particularly those which make use of three mentioned traditional methods. There is a decent amount of literature on this, which authors largely fail to acknowledge throughout the manuscript. I am left with no clue as to how your results compare to other results. This is a serious shortcoming of the manuscript.

Response 2:

Discussion and results were separated.

More references were used to help discuss the results

The Authors should fit the manuscript text and references lists to MDPI’s citations style based on the style used by the American Chemical Society (ACS). Numbers [1] instead name and year (Tasoulas et al., 2013). This needs to be corrected throughout the text.

Response 3:

Manuscript text and references were correct to fit MDPI’s citations style.

Summarizing review, this paper is unsuitable for publication in its current form. Authors should rewrite some parts of the work. Once these changes are made, this paper is a reasonable contribution to Forests Journal.

Reviewer 2 Report

General comment and remarks

The paper represents a valuable contribution to the topic. The language is good (one of the authors is native speaking I guess) and the methods are sound. The the results are interesting and pretty well described. I also found that the literature is pretty well known and many non-Chinese Authors are also cited, differently from other Chinese Authors which are not very prone to cite non-Chinese Authors. Anyway the main problem of the paper is that no discussion is provided, even if written as combined with the (valuable and sound) results. In its current form the paper can’t be accepted as research paper neither as a case study or technical note. Here attached the definition of what a Discussion section should be and what I expect to see in the updated version of the manuscript.

Specific comments

The reference are not formatted according to journal's guidelines

I suggest to create a unique Material and methods section a chapter 2 where the study area and the data and all the other sections (3. Methods, 3.1 Neighbours identification and so on) can be reported as sub-chapters.

You don’t need to write “see Figure xx” when referring to figures. This is implicit for a scientific paper.

A real discussion is mandatory

What a discussion chapter should include (from https://library.sacredheart.edu/c.php?g=29803&p=185933)

Definition

The purpose of the discussion is to interpret and describe the significance of your findings in light of what was already known about the research problem being investigated, and to explain any new understanding or fresh insights about the problem after you've taken the findings into consideration. The discussion will always connect to the introduction by way of the research questions or hypotheses you posed and the literature you reviewed, but it does not simply repeat or rearrange the introduction; the discussion should always explain how your study has moved the reader's understanding of the research problem forward from where you left them at the end of the introduction.

Importance of a Good Discussion

This section is often considered the most important part of a research paper because it most effectively demonstrates your ability as a researcher to think critically about an issue, to develop creative solutions to problems based on the findings, and to formulate a deeper, more profound understanding of the research problem you are studying.

The discussion section is where you explore the underlying meaning of your research, its possible implications in other areas of study, and the possible improvements that can be made in order to further develop the concerns of your research.

This is the section where you need to present the importance of your study and how it may be able to contribute to and/or fill existing gaps in the field. If appropriate, the discussion section is also where you state how the findings from your study revealed new gaps in the literature that had not been previously exposed or adequately described.

This part of the paper is not strictly governed by objective reporting of information but, rather, it is where you can engage in creative thinking about issues through evidence-based interpretation of findings. This is where you infuse your results with meaning.

Author Response

Response to Reviewer 2 Comments

General comment and remarks

The paper represents a valuable contribution to the topic. The language is good (one of the authors is native speaking I guess) and the methods are sound. The the results are interesting and pretty well described. I also found that the literature is pretty well known and many non-Chinese Authors are also cited, differently from other Chinese Authors which are not very prone to cite non-Chinese Authors. Anyway the main problem of the paper is that no discussion is provided, even if written as combined with the (valuable and sound) results. In its current form the paper can’t be accepted as research paper neither as a case study or technical note. Here attached the definition of what a Discussion section should be and what I expect to see in the updated version of the manuscript.

 Response 1:

Thanks a lot for the very helpful comments, Discussion section has been added to the article.

Specific comments

The reference are not formatted according to journal's guidelines

Response 2:

The references have been correct according to journal’s guidelines.

I suggest to create a unique Material and methods section a chapter 2 where the study area and the data and all the other sections (3. Methods, 3.1 Neighbours identification and so on) can be reported as sub-chapters.

Response 3:

“Material and methods” section was created for chapter 2, which included study area, data set, and methods three sub-chapters.

You don’t need to write “see Figure xx” when referring to figures. This is implicit for a scientific paper.

Response 4:

All the words referring to figures were removed from the article.

A real discussion is mandatory

What a discussion chapter should include (from https://library.sacredheart.edu/c.php?g=29803&p=185933)

Definition

The purpose of the discussion is to interpret and describe the significance of your findings in light of what was already known about the research problem being investigated, and to explain any new understanding or fresh insights about the problem after you've taken the findings into consideration. The discussion will always connect to the introduction by way of the research questions or hypotheses you posed and the literature you reviewed, but it does not simply repeat or rearrange the introduction; the discussion should always explain how your study has moved the reader's understanding of the research problem forward from where you left them at the end of the introduction.

Importance of a Good Discussion

This section is often considered the most important part of a research paper because it most effectively demonstrates your ability as a researcher to think critically about an issue, to develop creative solutions to problems based on the findings, and to formulate a deeper, more profound understanding of the research problem you are studying.

The discussion section is where you explore the underlying meaning of your research, its possible implications in other areas of study, and the possible improvements that can be made in order to further develop the concerns of your research.

This is the section where you need to present the importance of your study and how it may be able to contribute to and/or fill existing gaps in the field. If appropriate, the discussion section is also where you state how the findings from your study revealed new gaps in the literature that had not been previously exposed or adequately described.

This part of the paper is not strictly governed by objective reporting of information but, rather, it is where you can engage in creative thinking about issues through evidence-based interpretation of findings. This is where you infuse your results with meaning.

 Response 5:

Discussion section was added to the article.

Round 2

Reviewer 2 Report

Dear Authors, many thanks for providing this new version of the paper which seems to be really improved. The shortcoming of the discussion section has been solved. I just saw few things to be adjusted prior acceptance. The main thing I saw is still a discussion section which needs to be polished. Indeed it is still unclear whether the results you achieved are better or worse than the data you can find in literature. The discussion also needs to begin with a strong sentence accepting or rejecting you research question. In other words, I believe that the first 2 sentences should answer to the main question: which method was the best and why? Probably the lines 432-438 might be moved here...

As regard references please solve some errors. As for example on L58 the reference [17] should be Tseng et al. and not Tseng, Lin. The same with reference [8] L64 and other cases such as L71, L77, L80, L83. Please take care about that.

In my opinion the use of a single M&M section with sub-sections makes the paper much more professional and readable. thanks for accepting my suggestion. Just pay attention to format all the sub-sections titles with the same font/style.

Concerning figures I think a misunderstanding occurred. In the current version some figures are never cited in the text, e.g. Figure 3 or Figure 4. In my previous revision my aim was to highlight that you don't need to use the verb "see" in combination with "Figure x" but just to call the Figure. In the current version you need to adjust it again in order to guide the reader when he/she should direct his/hes eyes on the images. Then please avoid to call the same figure too several times. In general, just one at the beginning or the end of the relative paragraph. Then call the whole figure, not a subsection of it (e.g. 5a or 5b).

In conclusion I applaud the Authors for what they have done until now, the paper is full of interesting data and knowledge but I believe it still needs to be refined in the Discussion section
